# Cholera outbreak in Forcibly Displaced Myanmar National (FDMN) from a small population segment in Cox's Bazar, Bangladesh, 2019

Abu S. G. Faruque[1‡]*, Azharul Islam Khan[1‡], Baitun Nahar[1], S. M. Rafiqul Islam[1], M. Nasif Hossain[1], Syed Asif Abdullah[1], Soroar Hossain Khan[1], Md. Sabbir Hossain[2], Feroz Hayat Khan[2], Mukeshkumar Prajapati[2], Yulia Widiati[3], A. S. M. Mainul Hasan[3], Minjoon Kim[4], Jennie Musto[2], Maya Vandenent[4], John David Clemens[1,5], Tahmeed Ahmed[1]

1 Nutrition and Clinical Services Division, icddr,b, Dhaka, Bangladesh, 2 World Health Organization, Cox's Bazar, Bangladesh, 3 UNICEF Bangladesh, Cox's Bazar Field Office, Cox's Bazar, Bangladesh, 4 UNICEF Bangladesh Country Office, Cox's Bazar, Bangladesh, 5 UCLA Fielding School of Public Health, Los Angeles, California, United States of America

‡ These authors contributed equally to this work as joint first authors
* gfaruque@icddrb.org

**Data Availability Statement:** To protect patient confidentiality according to consent agreements,

## Abstract

### Background

Bangladesh experienced a sudden, large influx of forcibly displaced persons from Myanmar in August 2017. A cholera outbreak occurred in the displaced population during September-December 2019. This study aims to describe the epidemiologic characteristics of cholera patients who were hospitalized in diarrhea treatment centers (DTCs) and sought care from settlements of Forcibly Displaced Myanmar Nationals (FDMN) as well as host country nationals during the cholera outbreak.

### Methods

Diarrhea Treatment Center (DTC) based surveillance was carried out among the FDMN and host population in Teknaf and Leda DTCs hospitalized for cholera during September-December 2019.

### Results

During the study period, 147 individuals with cholera were hospitalized. The majority, 72% of patients reported to Leda DTC. Nearly 65% sought care from FDMN settlements. About 47% of the cholera individuals were children less than 5 years old and 42% were aged 15 years and more. Half of the cholera patients were females. FDMN often reported from Camp # 26 (45%), followed by Camp # 24 (36%), and Camp # 27 (12%). Eighty-two percent of the cholera patients reported watery diarrhea. Some or severe dehydration was observed in 65% of cholera individuals. Eighty-one percent of people with cholera received pre-

the policy of the data gathering centre (icddr,b) limits the public availability of the whole data set in the manuscript, the supplemental files, or a public repository. However, part of the data set related to this manuscript is available upon request and readers may contact with Ms. Armana Ahmed (aahmed@icddrb.org) of the Research Administration & Strategy of icddr,b to request the data (http://www.icddrb.org/).

**Funding:** The study was funded by UNICEF Bangladesh. Grant number: GR- 01875. The funders had no role in study design, data collection and analysis, decision to publish, or preparation of the manuscript.

**Competing interests:** The authors have declared that no competing interests exist.

packaged ORS at home. About 88% of FDMN cholera patients reported consumption of public tap water. Pit latrine without water seal was often used by FDMN cholera individuals (78%).

## Conclusion

Vigilance for cholera patients by routine surveillance, preparedness, and response readiness for surges and oral cholera vaccination campaigns can alleviate the threats of cholera.

## Author summary

Bangladesh observed a large-scale arrival of forcibly displaced individuals from Myanmar in August 2017. The Bangladesh Government, UN agencies, and international and national non-governmental organizations responded immediately with extensive humanitarian response. However, threats of cholera outbreaks were prevailing. The Government of Bangladesh as lead, with technical support from icddr,b collaborating with international agencies undertook a massive oral cholera vaccination (OCV) campaign immediately as a pre-emptive measure to alleviate threats of the cholera outbreak. Despite that mass OCV campaign, threats of cholera outbreak were existing due to new arrivals of the displaced population with compromised host susceptibility, frequent visits to settlements by Bangladesh nationals without exposure to OCV, and the declining vaccine immunity among OCV recipients as well as an increasing number of cohort children without any exposure to OCV. The population faced a cholera outbreak during September-December 2019. This study aims to describe the characteristics of cholera patients, their care-seeking pattern, camp-wise distribution, source of drinking water, sanitation facility, OCV status, and share the experiences from effective interventions to prevent a cholera outbreak. Vigilance for cholera patients by routine surveillance, preparedness for both preventive and control measures, and response readiness for surges and OCV campaigns can alleviate the threats of cholera.

## Introduction

In August 2017, Bangladesh witnessed a sudden influx of an estimated over 700,000 Forcibly Displaced Myanmar Nationals (FDMN) including large number of children within shortest possible time from neighboring Rakhine state in Myanmar who settled in the Cox's Bazar district situated in the south-east of the country [1–4]. The Bangladesh Government, UN agencies, and a large number of international and national non-governmental organizations (NGOs) reacted immediately with a large-scale humanitarian response. Camps were established quickly but soon humanitarian agencies started struggling to meet the exorbitant demand for assistance and supplies [1,2,5,6]. The displaced population urgently needed critical supplies like medicine, clean water, food, and shelter with special attention to children, women, the elderly, and disabled individuals [2,3,6]. Many of the hurriedly built camps were vulnerable to monsoon flooding and storm surges. Those families who started living in hillsides were prone to landslides. Latrines and shallow and deep tube wells were constructed to protect against public health issues and ensure access to clean water [3,4,7]. However, because of the arrival of a large number of displaced populations and the presence of insufficient

lifesaving infrastructures of sanitation, like latrines and waterpoints, the environment soon became a breeding place for waterborne diseases including acute watery diarrhea, cholera, and shigellosis [2,3,5]. These risks were further heightened by high population density in camps and an excess number of severely malnourished children who often yield more quickly to preventable and treatable diseases as well as outbreaks of acute watery diarrhea (AWD), cholera, and shigellosis [2–4,6].

Almost immediately, following the huge influx and settlement of these displaced populations, UNICEF-Bangladesh and icddr,b jointly conducted a brief field assessment in the Ukhia and Teknaf sub-districts of Cox's Bazar. The assessment anticipated potential threats of diarrheal disease outbreaks including cholera and shigellosis, and strategies were immediately set to initiate mitigation measures. A partnership between icddr,b, and UNICEF under the umbrella of Health Sector targeted (i) training doctors, nurses, and community health workers of the government and NGO run facilities serving FDMN in the settlements as well as host population living in the neighborhood housing; (ii) managing people with dehydrating diarrheal episodes and associated malnutrition through a network of five diarrhea treatment centers (DTCs); and (iii) carrying out DTC based diarrheal disease surveillance as it is known to be critical for early detection of outbreaks. Activities of the diarrheal disease surveillance team included data collection, a one-step rapid diagnostic test for the presence of *Vibrio cholerae* in stool specimen of hospitalized patients, and microbial tests to detect common enteric pathogens including *Vibrio cholerae* by submitting fecal specimens directly as well as after inoculation into Cary-Blair Transport Medium to the Clinical Microbiology Laboratory of icddr,b in Dhaka, Bangladesh.

The Government of Bangladesh as lead, with technical support from icddr,b collaborating with international agencies, and international and national NGOs under the wider platform of Health Sector, undertook a massive oral cholera vaccination (OCV) campaign immediately as a pre-emptive measure to alleviate threats of cholera outbreaks [8–10]. Despite that mass OCV campaign, threats of cholera outbreaks among FDMN were existing due to new arrivals of the displaced population with compromised host susceptibility, frequent visits to settlements by Bangladesh nationals living in the neighboring community without exposure to OCV, as well as an increasing number of cohort children without any exposure to OCV. Preparedness for preventive and control measures to combat surges and vigilance for people with cholera were the most important public health priorities because of prevailing threats of cholera in both the host and displaced population [11,12].

icddr,b, and UNICEF jointly organized a dissemination session for the local stakeholders on their activities for the FDMN living in the settlements in March 2019. Between September and December 2019, there have been 147 people with culture-confirmed cholera who presented and subsequently hospitalized with acute dehydrating diarrhea episodes in Leda and Teknaf DTCs. Thus, it became essential to share this cholera outbreak control experience with policymakers, public health teams, program managers, academia, and wider stakeholders acquired from a strategy in an emergency and crisis setting. Such experience sharing is not a common and widespread phenomenon, particularly in humanitarian emergencies. An update of this kind is likely to enable stakeholders to undertake necessary preparedness to prevent cholera outbreaks from occurring and to respond successfully when the outbreaks have occurred.

In late September 2019, two cholera patients for the first time after two years of the arrival of FDMN were detected in Teknaf DTC which is run by icddr,b. They sought care from settlements (one from Camp # 25 and the other from Camp # 26). Such an incident was reported immediately to the Epidemiology Team Lead and Early Warning, Alert and Response System (EWARS) of WHO-Cox's Bazar, as well as UNICEF-Cox's Bazar. The next day, Cox's Bazar

Health Sector's Joint Assessment Team (JAT) consisting of Health and WASH Sector partners investigated the hotspots and affected camps. The JAT reported worsening hygiene practices and sanitary conditions as a result of an acute shortage of safe drinking water, and the use of stagnant contaminated water for domestic purposes. Several recommendations were made on that day including hygiene promotion in the hot spots, desludging of latrines as soon as possible, distribution of water purifying tablets, pre-packaged ORS, soap, and chlorine by the WASH Sector, and availability of a handwashing facility in the latrine areas. The stagnant contaminated pools of water were fenced to prevent access to it by people living in its surroundings. Urgent refresher training on risk assessment for health teams was recommended. Within 24 hours, one temporarily closed down DTC in Leda nearby the affected settlements was reopened to serve the increasing number of AWD patients.

The Health Systems of Bangladesh Government continued collaboration with WHO-Cox's Bazar in streamlining activities of EWARS, actively involved in strengthened monitoring of the individuals with AWD and cholera in the camps for early detection and response to outbreaks. Institution of immediate alleviation measures included the supply of safe drinking water and improvement of the sanitation system. To ensure adequate clinical management of AWD individuals following a standard management protocol, the existing network of DTCs was strengthened by UNICEF-Cox's Bazar. WHO and the Health Sector recommended that those patients presenting to the out-patient clinics with dehydrating diarrhea should be immediately referred to Diarrhea Treatment Centres (DTCs) run by icddr,b, or, if there were no DTCs nearby, to primary health care centers (PHCs) with isolation facilities. Leda DTC (14 beds) and Teknaf DTC (30 beds) located in the neighborhood of settlements remained open as usual round-the-clock. Six batches of the health workforce were immediately trained by icddr,b on the clinical management of AWD individuals. Community health workers were also assigned by UNICEF-Cox's Bazar in outreach activities including promotion of good hygiene practices and combatting diarrhea episodes at the household level with the use of pre-packaged ORS as soon as there was onset of these episodes [13–16].

Preparations and response readiness were undertaken for the acceleration of the existing cholera vaccination campaign as an increasing trend of dehydrating diarrhea patients in DTCs was revealed. As a result, the International Coordinating Group for Cholera Vaccine (ICG) Secretariat approved a request for additional 1.2 million doses of OCV. Ministry of Health and Family Welfare, Bangladesh playing the leading role with the support of WHO, UNICEF, and other partners, the campaign started vaccinating those individuals living in the neighborhood host community but yet to receive any OCV. The OCV campaign (including operational cost) was funded by GAVI, the Vaccine Alliance. The vaccination operation aimed mostly to reach displaced children aged 12–59 months. In the host community, the campaign looked for any person aged 1 year or more, because approximately 80% of host community people residing near the settlements were never targeted to receive OCV in previous campaigns although they were equally vulnerable like the FDMN [13,14].

This paper aims to (i) describe the characteristics of cholera patients including that of FDMN care seekers, their reporting pattern to DTCs, camp-wise distribution, and OCV status, (ii) compare drinking water sources and toilet use pattern between FDMN and host community cholera individuals, (iii) describe comparative clinical and demographic characteristics between cholera individuals who sought care from Cox's Bazar DTCs, and Dhaka Hospital of icddr,b during the same period, and (iv) share the experiences that were obtained from this cholera outbreak that occurred in a small segment of the FDMN living in settlements of Cox's Bazar, Bangladesh.

## Methods

### Ethics statement

The data collection process of this study was part of the ongoing activities entitled: *Surveillance for etiologic agents*, *care-seeking behavior, the status of IYCF and WASH practices among patients or their caregivers from Rohingya refugees as well as host population in Cox's Bazar district attending icddr,b operated Diarrhea Treatment Centers* was approved by icddr,b's (International Centre for Diarrhoeal Disease Research, Bangladesh) IRB (PR-17111; December 5, 2017) comprising Research Review Committee (RRC) and Ethical Review Committee (ERC). Voluntary informed written consent was obtained from the parent/guardian before starting the interviewing process.

### Setting and study population

This was a DTC-based cross-sectional diarrheal disease surveillance for FDMN and host community individuals hospitalized in DTCs located in Leda and Teknaf from September to December 2019.

### Stool sample collection, rapid diagnostic testing, and laboratory methods

Routine enteric pathogen detection activities that included a collection of a single stool specimen (of at least 3 g) directly from the patients following hospitalization were ongoing in DTCs. Soon after collection, a one-step rapid diagnostic test was performed by SD BIOLINE cholera antigen O1/O139 (44FK30) test kit, supplied by WHO-Cox's Bazar, which is an immunochromatographic test for the qualitative detection of *Vibrio cholerae* O1/O139 in human stool specimens (manufactured by STANDARD DIAGNOSTICS, INC located in Suwon city, Kyonggi province, Republic of Korea). To facilitate microbial culture to confirm the rapid diagnostic test results; the provisionally diagnosed specimens (the stool) of cholera patients were inoculated into Cary-Blair Transport Medium; and the medium was then sent as soon as possible to the Clinical Microbiology Laboratory, icddr,b, based in Dhaka, Bangladesh to isolate the colony as well as perform antibiotic susceptibility tests with immediate sharing of the results to the concerned DTC, Epidemiology Team Lead of WHO-Cox's Bazar and UNICEF-Cox's Bazar. Other non-positive by rapid diagnostic test specimens were submitted routinely once or twice a week [17–19].

### Data collection

In daily monitoring, evaluation, and reporting, the present study followed DTC based diarrheal disease surveillance system (DDSS) in Teknaf and Leda for culture confirmed cholera patients during September-December 2019. Ongoing data collection by trained research assistants entailed administering structured questionnaires, from all hospitalized patients in DTCs and/or their attendants to gather information such as presenting clinical features, socioeconomic and demographic contexts, water, sanitation and hygiene, housing and its surrounding environment, feeding practices, particularly of 0–35 months old, and use of drugs and prepackaged ORS at home before coming to DTCs that continued serving round-the-clock. During the interview of host population, research assistants were comfortable with the native Bengali language; however, when needed particularly in case of FDMN they received assistance of DTC staff members who understood the dialect of FDMN and was familiar with their culture, day to day living patterns and housing environments in settlements.

### Statistical analysis

Data were analyzed by STATA (StataCorp version 13) and analyses included descriptive methods. Variables were described using frequencies with percentages. Exposure categories were compared using the Chi Square test for categorical variables. Relevant data from the ongoing DDSS database of Dhaka Hospital were extracted for the period September-December, 2019 for a comparative analysis of clinical and demographic profiles of visiting culture-proven cholera patients between Cox's Bazar DTCs and Dhaka Hospital of icddr,b.

## Results

Between September and December 2019, there were 147 culture-confirmed cholera patients presented and were subsequently hospitalized with acute dehydrating diarrhea episodes in Leda and Teknaf DTCs. The majority, 72% of cholera individuals reported to Leda DTC. Nearly 65% of these cholera patients sought care from FDMN settlements. FDMN often reported to DTCs from Camp # 26 (45%), followed by Camp # 24 (36%), and Camp # 27 (12%). About 94% of the cholera patients from the host community and 65% of the cholera individuals from FDMN living in settlements did not receive any OCV before their onset of culture-proven cholera episodes (Table 1). Overall, these DTCs served during the outbreak an estimated 22% of both FDMN living in settlements and host country nationals residing in the neighborhood (Fig 1).

The major sources of drinking water of the hospitalized displaced cholera individuals were public tap installed in the settlements, deep tube-well, and shallow tube well. Use of public tap water was significantly more frequent in cholera patients from settlements than from the host community (88% vs. 10%; p<0.001). However, the use of deep tube well (6% vs. 21%; p = 0.005) and shallow tube well (2% vs. 54%; p<0.001) water was significantly less common in the cholera patients from settlements. Nearly 78% of the displaced cholera patients used pit latrines without water seal as opposed to 44% of the individuals with cholera from the host

**Table 1. Distribution of characteristics of culture-confirmed cholera patients (n = 147) in Leda and Teknaf DTCs in Cox's Bazar settlements, September-December 2019.**

| Variables name | n (%) |
|---|---|
| **Sought care from** | |
| Leda DTC | 106 (72.1) |
| Teknaf | 41 (27.9) |
| **Currently living in** | |
| Settlements | 95 (64.6) |
| Host community | 52 (35.4) |
| **From settlements** | |
| Camp # 26 | 43 (45.3) |
| Camp # 24 | 34 (35.8) |
| Camp # 27 | 11 (11.6) |
| Camp # 25 | 4 (4.2) |
| Camp # 15 | 2 (2.1) |
| Camp # 23 | 1 (1.1) |
| **Not exposed to OCV** | |
| FDMN | 62 (65.3) |
| Host community individuals | 49 (94.2) |

DTC: Diarrhea treatment center; OCV: Oral cholera vaccine

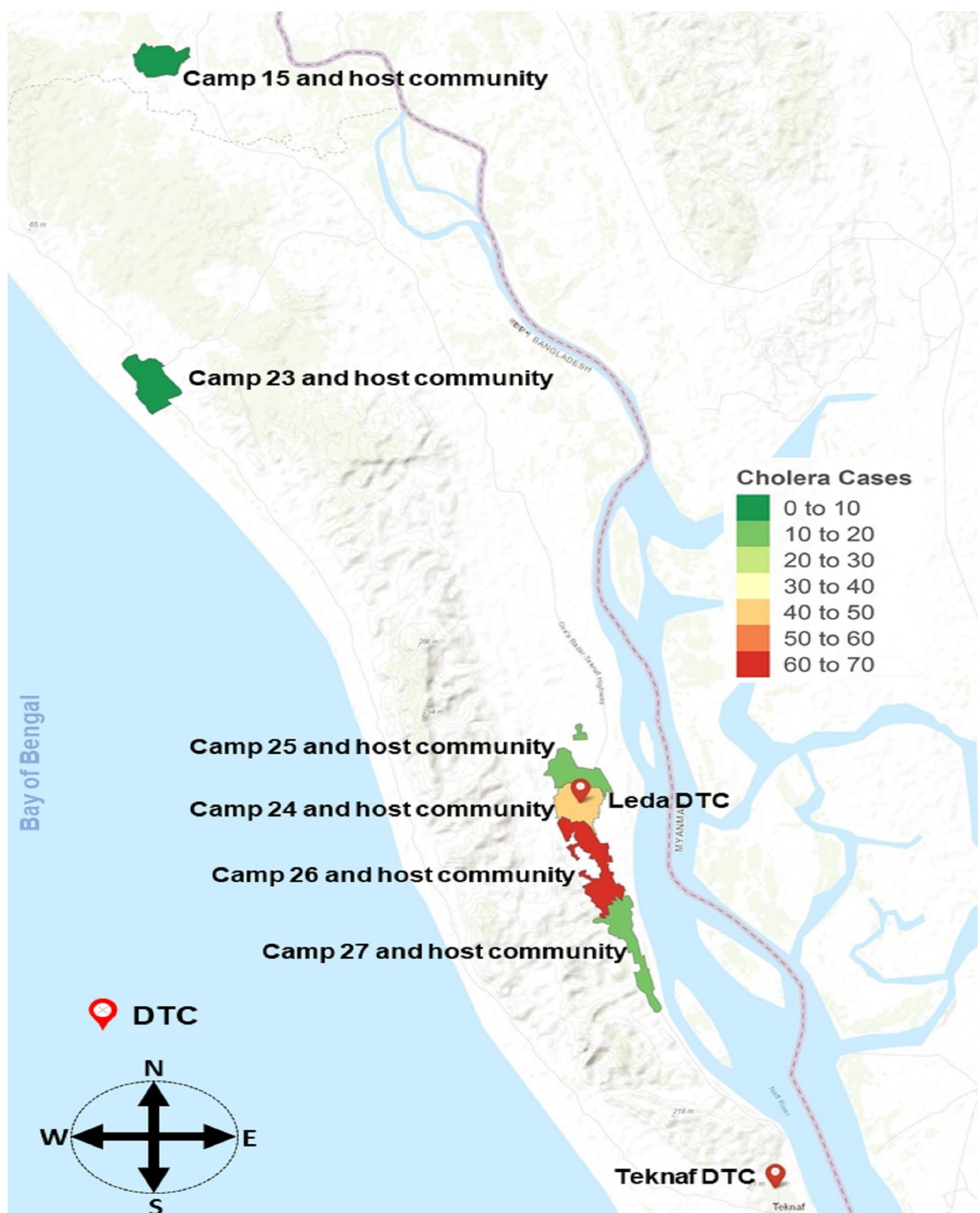

**Fig 1. Cholera detected region, Leda DTC and Teknaf DTC.** The map is generated using R version 4.0.2 with Esri—Esri, DeLorme, NAVTEQ.

community (p<0.001). However, the use of a pit latrine with a water seal was identical in both the groups (Table 2).

During September-December 2019, a total of 216 culture-confirmed cholera individuals were hospitalized in icddr,b's Dhaka Hospital, and none had received OCV. During the same period, DTC logs reported the admission of 147 culture-proven cholera patients in Leda and Teknaf DTCs. Among these cholera patients, infants (p<0.001) and overall children <5 years old (p<0.001) presented more frequently to the DTCs (functioning to treat FDMN living in settlements as well as host community individuals) compared to cholera children presenting to Dhaka Hospital from Dhaka city and its suburbs (47% vs. 12%; p<0.001). However, for individuals aged 15 years and higher, more cholera patients reported to Dhaka Hospital as opposed to cholera patients living in settlements and seeking care from DTCs (76% vs. 42%; p<0.001) (Table 3). Significantly more female cholera patients visited DTCs as opposed to female cholera patients presenting to Dhaka Hospital (50% vs. 38%, p<0.043). People with cholera in Dhaka Hospital more commonly presented with watery diarrhea than cholera patients of DTCs (100% vs. 82%, p<0.001), sought care more frequently with some or severe dehydration (98% vs. 65%, p<0.001), and had more access to ORS at home before seeking care (91% vs. 81%, p<0.010) (Table 3).

## Discussion

Humanitarian emergencies increase the risk of infectious disease transmission including cholera and shigellosis, and the prevalence of other health conditions such as severe undernutrition. In a given similar scenario with preparedness for both preventive and control measures and response readiness, our observations highlighted the vital role of an effective disease surveillance system that continually generates essential epidemiologic data for effective strategy formulation. Such a system is critical for early detection of disease outbreaks before any spread to other family members as well as individuals living in the neighborhood, unnecessarily costing lives and challenging the disease control efforts. Thus, our ongoing DTC-based diarrheal disease surveillance system with timely laboratory back-up and immediate reporting to all concerned agencies was noteworthy in this emergency and crisis setting. The surveillance system was involved not only in collecting reliable data since the inception of the DTC network but also in reporting immediately to help significantly in anticipating and detecting early potential

**Table 2. Water source and toilet use by the culture-confirmed cholera patients in Leda and Teknaf DTCs in Cox's Bazar settlements, September-December 2019.**

| Variables | FDMN<br>*n = 95* (%) | Host community<br>*n = 52* (%) | P-value |
|---|---|---|---|
| **Water source** | | | |
| Public tap | 84 (88.4) | 5 (9.6) | <0.001 |
| Deep tube well | 6 (6.3) | 11 (21.2) | 0.005 |
| Shallow tube well | 2 (2.1) | 28 (53.8) | <0.001 |
| Others | 3 (3.2) | 8 (15.4) | 0.005 |
| **Toilet use pattern** | | | |
| Pit latrine without water seal | 74 (77.9) | 23 (44.2) | <0.001 |
| Pit latrine with water seal | 21 (22.1) | 12 (23.1) | 0.819 |
| Others | 0 (0.0) | 17 (32.7) | <0.001 |

**Table 3. Age stratified cholera patients in Dhaka Hospital and DTCs in Cox's Bazar settlements, September-December 2019.**

| Variables | Dhaka hospital n = 216 (%) | DTCs in settlements n = 147 (%) | p-value |
|---|---|---|---|
| **Age (Year)** | | | |
| <1 | 3 (1.4) | 14 (9.5) | <0.001 |
| <5 | 25 (11.6) | 69 (46.9) | <0.001 |
| 5–14 | 28 (13.0) | 17 (11.6) | 0.814 |
| 15 and more | 163 (75.5) | 61 (41.5) | <0.001 |
| **Range** | 7 months– 74 years | 3 months– 85 years | |
| **Female** | 83 (38.4) | 73 (49.7) | 0.043 |
| **Duration of diarrhea** | | | |
| <1 day | 153 (70.8) | 100 (68.0) | 0.300 |
| 1–3 days | 57 (26.4) | 38 (25.9) | 0.994 |
| 4 days and more | 6 (2.8) | 9 (6.1) | 0.192 |
| **Watery stool** | 216 (100.0) | 120 (81.6) | <0.001 |
| **Some or severe dehydration** | 211 (97.7) | 95 (64.6) | <0.001 |
| **Pre-packaged ORS us at home** | 196 (90.7) | 119 (81.0) | 0.010 |

DTC: Diarrhea treatment center; ORS: Oral rehydration solution

cholera outbreaks. Findings from surveillance system guided intervention strategies that led to the timely undertaking of preventive measures and the preparedness that included training of health care staff, opening of temporarily closed down DTC, strengthening of existing DTCs, outreach activities, and prepositioning of supplies as well as additional human resources. Other additional vital strategies undertaken were inter-sectoral collaboration, strengthening of preventive and control measures (regular monitoring of the quality of drinking water sources at waterpoints and household level, sanitation as well hygiene) as well as OCV campaigns. Efforts further emphasized preparedness for surges and vigilance for cholera patients which was the priority undertakings of the Health Systems of the Government of Bangladesh because of existing threats of cholera in both the host and displaced populations in emergency and crisis settings.

Additionally, surveillance data helped in identifying vulnerable populations living in high-risk areas who might have been benefitted from preventive OCV use. Thus, reliable epidemiological data was critical in the efficient implementation of preventive as well as control measures.

The present study observed that 94% of the host community individuals and two-third of the FDMN with laboratory-confirmed cholera were not exposed to OCV before getting hospitalized with AWD. A recent experience from Bangladesh and India indicated that the protective efficacy of Shanchol OCV (produced in India) among those more than five years against cholera is 53–65%. The study mentioned the positive role of OCV as a pre-emptive measure in endemic settings, in natural or man-made disasters even in disruptive situations with a breakdown of WASH and absence of other disease control and public health measures [20]. WHO and Global Task Force for Cholera Control (GTFCC) recommend that a comprehensive multi-sectoral involvement is important for the successful elimination of cholera [21]. Mass OCV campaigns with high coverage are feasible even after the arrival of a large number of displaced populations in a distressed state in resource poor settings like Bangladesh [8,9]. According to another study, OCV induced optimal immune responses in FDMN adults and children which were similar to that observed in Bangladesh's population of diverse age groups or individuals living in other cholera endemic countries [10]. In Sudan among the displaced

populations, the risks for cholera were considerably higher among children less than five years living in refugee camps [22]. A Cochrane review indicated significantly lower protective efficiency of OCV in under-five children compared to children who are older than them as well as adults [23]. Vigilance for cholera individuals as well as preparedness for prevention and mitigation measures for surges and mass OCV campaigns for FDMN as well as host population can reduce the threats of cholera in both the host and FDMN [24–29].

In this study, we have explored the clinical, demographic, and hygienic practices of the displaced as well as the host population living in settlements and neighboring host communities. The findings of this study have public health implications and may be useful for the Health System of the Government of Bangladesh for anticipation, preparedness, and implementation of preventive and mitigation measures in settings with public health threats such as endemic disease surges like cholera or it is breaking out into epidemic proportions. Additionally, vigilance for cause-specific diarrhea surges in both the populations such as host and FDMN is critical. Several findings related to care-seeking from DTCs were noteworthy. Unlike Dhaka hospital, children living in settlements and host communities were more often hospitalized for culture-proven cholera episodes than their peers from Dhaka city and its suburbs. These observations underscore the need for OCV campaigns. Females aged 15 years and higher living in settlements were more often hospitalized with cholera than their peers seeking care from Dhaka Hospital. This may be due to the increased vulnerability of females living in settlements to cholera because of their higher compromised immunity or excess exposure to contaminated water and food during household activities. Excess reporting of male cholera patients in Dhaka Hospital may be due to increased mobility of male individuals as well as their frequent exposures to day-time unhygienic outdoor street-side meals or snacks from vendors in the overcrowded megacity.

ORS use at home was significantly lower in the cholera patients seeking care from DTCs than those cholera individuals living in Dhaka city and its suburbs. A big factor limiting people's use of ORS is their knowledge of when and how to use this vital tool. Major limitations of outreach activities in this scenario may include less promotion and access to ORS packets at the household or community level in settlements, because of less organized outreach activities. Additionally, lack of appropriate health education measures to make FDMN knowledgeable about ORS use particularly when to start, how to prepare, how much to be taken, and how long to be continued. All these more effective attempts may motivate FDMN to enhance their appropriate use of ORS at the household level before coming to DTCs.

Access to more safe water (chlorinated water supplied through taps installed) was observed in settlements mostly for FDMN as provided by international agencies and NGOs. However, their access to deep and shallow tube well water was less commonly observed compared to that of admissions from the host community. It is important that treatment of water is a vital tool for providing safe water when tube wells are inadequate in meeting the needs of the displaced population in emergency and crisis settings.

Cholera patients with significantly more frequent watery stool and with more common evidence of some or severe dehydration in Dhaka hospital could be due to more full-blown clinical features of cholera episodes which may be because of larger inoculum size that may be ingested by those living in the more contaminated environment particularly in slums with gross lack of water and sanitation services as well as worsening hygienic practices in Dhaka city and its suburbs.

icddr,b followed its expertise gathered from its hospital-based Diarrheal Disease Surveillance System (DDSS) which is in operation in icddr,b's urban Dhaka (since 1979), and rural Matlab (since 1999) facilities. The Diarrheal Disease Surveillance System (DDSS) at Dhaka Hospital enrolls a 2% systematic sample of patients reporting to the triage area. Patients

seeking care from the Matlab Hospital who are residents of the Health and Demographic Surveillance System (HDSS) area are enrolled into the DDSS. Trained enumerators using structured questionnaires interview patients and/or their attendants to collect relevant information. Microbiological assessments are performed to identify common diarrheal pathogens and document the microbial susceptibility pattern of the bacterial pathogens. The activity offers useful information to hospital clinicians in their clinical decision-making courses and empowers icddr,b to detect the emergence of new enteric pathogens and early recognition of outbreaks and their locations, thereby guiding the host government to take suitable preventive and control measures [17–19].

There was an absence of comparable diarrhea treatment facilities in the settlements which not only providing quality care but also examining stool specimens for diarrheagenic organisms following standard laboratory methods. We needed data for comparison of presenting clinical and demographic features of hospitalized cholera patients (such as age, sex, duration of diarrhea, watery stool, dehydration status, and pre-packaged ORS use) from Leda and Teknaf DTCs with that of a facility that has a track record of diarrheal disease surveillance system and treating hospitalized cholera patients who are seeking care from such a facility that does not charge for the services, provides quality care mostly to those attending from poor socio-economic contexts, remains open round-the-clock, and can efficiently handle sudden upsurges of patients including individuals with cholera presenting often in a dehydrated state in a relatively large number and the facility has a back-up laboratory for routine fecal specimen examinations following standard methods for detection and characterization of causative enteric organisms including *V. cholerae*.

This study has few limitations and one of the limitations was these activities were DTC based as a result only those cholera individuals with admissions in DTCs have been included in the study. Cholera patients with less severe disease who reported to the DTCs and received care on an outpatient basis for a brief period and those patients who developed cholera at the community and did not report to DTCs have not been studied. Thus, results may not be generalizable. However, the study of a fairly large number of cholera patients captured during an outbreak as well as quality laboratory performance were the strengths of the study.

## Conclusion

Threats of cholera outbreaks among the FDMN are continuing due to new arrivals with compromised host susceptibility, as well as an increasing number of cohort children without any exposure to OCV. Quality surveillance and rapid microbial confirmation of provisionally diagnosed suspected individuals with cholera have important public health implications in emergencies and crises. Continued preventive and control measures, preparedness and response readiness for surges, and vigilance for cholera patients should be the priority undertakings of the Health Systems of the Government of Bangladesh because of existing threats of cholera in both the host and displaced populations.

## Supporting information

**S1 STROBE checklist. STROBE checklist of items for *cross-sectional studies*.**
(DOCX)

**S1 Data. Region-wise cholera cases.** This dataset contain number of cholera cases settlement and host population visit to two DTC (Leda and Teknaf) form several region.
(CSV)

**S2 Data. This dataset contain exact DTC location with with latitude and longitude.**
(CSV)

## Acknowledgments

We acknowledge the contribution of icddr,b's core donors including the Government of the People's Republic of Bangladesh, Global Affairs Canada, Canada; Swedish International Development Cooperation Agency, and UK Aid (FCDO) for their continuous support and commitment to the icddr,b's research efforts.

## Author Contributions

**Conceptualization:** Abu S. G. Faruque, Tahmeed Ahmed.

**Data curation:** Abu S. G. Faruque, M. Nasif Hossain, Soroar Hossain Khan.

**Formal analysis:** M. Nasif Hossain, Soroar Hossain Khan.

**Investigation:** Abu S. G. Faruque, Azharul Islam Khan, Tahmeed Ahmed.

**Methodology:** Abu S. G. Faruque, Azharul Islam Khan, S. M. Rafiqul Islam, M. Nasif Hossain, Soroar Hossain Khan.

**Project administration:** Abu S. G. Faruque, S. M. Rafiqul Islam.

**Supervision:** Abu S. G. Faruque, M. Nasif Hossain.

**Writing – original draft:** Abu S. G. Faruque, Azharul Islam Khan, S. M. Rafiqul Islam, M. Nasif Hossain, Syed Asif Abdullah, Soroar Hossain Khan.

**Writing – review & editing:** Abu S. G. Faruque, Azharul Islam Khan, Baitun Nahar, S. M. Rafiqul Islam, Syed Asif Abdullah, Soroar Hossain Khan, Md. Sabbir Hossain, Feroz Hayat Khan, Mukeshkumar Prajapati, Yulia Widiati, A. S. M. Mainul Hasan, Minjoon Kim, Jennie Musto, Maya Vandenent, John David Clemens, Tahmeed Ahmed.

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
