## [Decision Letter · Decision Letter 0]

25 Mar 2021

Dear Dr. Abu S. G. Faruque,

Thank you very much for submitting your manuscript "Cholera outbreak in Forcibly Displaced Myanmar National (FDMN) from a small population segment in Cox’s Bazar, Bangladesh, 2019" for consideration at PLOS Neglected Tropical Diseases. As with all papers reviewed by the journal, your manuscript was reviewed by members of the editorial board and by several independent reviewers. In light of the reviews (below this email), we would like to invite the resubmission of a significantly-revised version that takes into account the reviewers' comments. 

Please refer to the constructive feedback recommending a light re-organization of the information presented in this article, in part to enhance clarity on the broader significance of these results. Please elaborate on what this study may tell us that other studies of cholera in refugee camps have not already established and how the results of the study may inform policies and programs for this population? 

We cannot make any decision about publication until we have seen the revised manuscript and your response to the reviewers' comments. Your revised manuscript is also likely to be sent to reviewers for further evaluation.

Sincerely,

Vasantha kumari Neela

Associate Editor

Amy Gilbert

Deputy Editor

Reviewer's Responses to Questions

**Key Review Criteria Required for Acceptance?**

**Methods**

-Are the objectives of the study clearly articulated with a clear testable hypothesis stated?

-Is the study design appropriate to address the stated objectives?

-Is the population clearly described and appropriate for the hypothesis being tested?

-Is the sample size sufficient to ensure adequate power to address the hypothesis being tested?

-Were correct statistical analysis used to support conclusions?

-Are there concerns about ethical or regulatory requirements being met?

Reviewer #1: Overall, the objectives were clearly stated in the background section of the manuscript. However, I found the objectives, as written in the abstract, to be too long and comprehensive to follow. I would suggest shortening the objectives to a simple statement about collecting cholera epidemiology in a sample of refugees and host country nationals. 

The study design was appropriate for these objectives.

I found the methods section to be a little confusing. The first half (lines 139-185) of the methods appeared to me more of a background about the outbreak than the specific methods of the study. I strongly suggest that portions of that section be moved to the background, with others (how did they respond to the outbreak) moved to the conclusion. That way the results are framed around those two pieces of context. 

Further, this first half of the methods had a sub-header (setting and study population), but the rest of the methods section did not. I suggest including additional headers to help guide the reader and follow what to me was a very complex section of the paper. These could include description of the surveillance systems, survey and lab methods, statistical methods, and ethical review. 

I have some specific comments/questions:

-Why was Dhaka used as a comparison sample rather than a nearby clinic? One would expect patients in the high density urban center of Bangladesh to be very different from refugees living in camps. I applaud the inclusion of host nationals living in nearby settlements, but do not understand the inclusion of this comparator group. 

-Some additional context on the number of camps, number of DTCs serving those camps, and how far they are from Dhaka would be helpful. As someone with limited knowledge of Bangladesh, this context would help me understand the context of the results. 

-Which DTCs were included in the study? I believe this was stated somewhere, but it was lost in the massive amount of background information included in the methods. 

-Who was invited to the survey? Suspected cases, lab confirmed cases or both? 

-Over what period of time were data collected? This was mentioned in the background and results, but it should be included in the methods.

-What language were the surveys conducted in and who exactly conducted the surveys? Were they trained?

Reviewer #2: The article is sound on it's methodology, objectives of the study clearly articulated.

The study design is appropriate to address the stated objectives.The population is clearly described and appropriate.

The correct statistical analysis were used to support the conclusion and the concerns about ethical or regulatory requirements were met.

Reviewer #3: -The objectives of the study were very clear

-and the study design was appropriate to address the objectives of the study

- The population was clearly described and the sample was sufficient for the study objectives

-Correct statistical analysis was undertaken, but

**Results**

-Does the analysis presented match the analysis plan?

-Are the results clearly and completely presented?

-Are the figures (Tables, Images) of sufficient quality for clarity?

Reviewer #1: The results were well presented and matched the analysis plan. Appropriate statistical tests were run.

Tables 2 and 3 included some p values, but not others (which were mentioned in the narrative). I would suggest included all p-values in all tables. 

What proportion of all camps served by these DTCs were affected? I see that 6 camps have data, but how many camps were served?

Reviewer #2: Yes the analysis were appropriately done.And the data were clearly presented. However there is no graphical presentation of data. Adding graphical presentation or images would be nice.

Reviewer #3: -A careful analysis was presented that matched the analysis plan

-The results were clear and tables were of sufficient quality

**Conclusions**

-Are the conclusions supported by the data presented?

-Are the limitations of analysis clearly described?

-Do the authors discuss how these data can be helpful to advance our understanding of the topic under study?

-Is public health relevance addressed?

Reviewer #1: The conclusions are supported by the data and the limitations are described. The importance of OVC was well established, but the other findings were less well discussed. 

I am wondering about the significance of these results. What does this study tell us that other studies of cholera in refugee camps have not already established? How did these results inform policies and programs for this population? 

Structurally, I found the authors jumped around a bit too much. They started by discussing general findings, then specific issues around OVC. Then they turned to a focus on water, sanitation and ORS, before returning to OVC in Sudan. Why not include all discussion of OVC together in one place? I found this difficult to follow and parse out the main points.

In the section on ORS use, I think the authors miss a big point. They highlight the role that limited access to ORS plays, but then they explain that the refugee population might not be motivated to use ORS. What about knowledge? A big factor limiting people's use of ORS is their knowledge of when and how to use this vital tool. This appears to be blaming the victim rather than focusing on the limitations of the system in which they live. Further, no mention of water treatment is made, which in refugee camps is a vital tool to providing safe water when wells are inadequate.

Reviewer #2: Yes the conclusions are on the basis of study findings. The discussions are adequate and well argued with evidence.

The study is unique in two ways : one, it is about Cholera outbreak which is a public health emergency. Two, the population is forcibly displaced vulnerable group. The underlying cross-cutting issues are well discussed.

Reviewer #3: -The conclusions are supported by the data and limitations clearly described. 

-Authors have discussed how the study health public health understanding AWDs in humanitarian crisis as well as the public health relevance of the study.

**Editorial and Data Presentation Modifications?**

Reviewer #1: One issue I had with this manuscript was the use of the term case rather than people or patients. Towards the end of the results the terms patient or female/child case were used, which is an improvement because it humanizes this population. This is already a highly vulnerable population and reducing them down to a non-human cases is unnecessary and potentially harmful. I would suggest the more humanistic term and to standardize the term throughout. 

There were other minor grammatical and editorial issues I noted throughout (see attached)

Finally, see my previous comments about the organization of the paper. Much of the methods I feel could be moved to the background and again to the discussion. That way the results are framed around the beginning of the cholera outbreak and how the group initially responded, and then how they used these surveillance systems and results to inform programming and policy.

Reviewer #2: Minor revision

Reviewer #3: Minor revisions

- Authors need to include ethical approval number in the ethical statement. They also need to bring the ethical statement at the start of the methods section. 

Major revision

- A map of the study setting showing camps were patients originated and locations of the treatment centers would highly enriched this study. Please see my comments in the paper.

**Summary and General Comments**

Reviewer #1: Overall, this paper presents novel data about a cholera outbreak amongst Myanmar refugees in Bangladesh, highlighting the important role that OVC plays in preventing disease. It also discusses the demographic make up and health seeking behaviors of this population. However, I am left wondering what the significance is. How does this advance the literature of cholera in refugee populations in general, and specifically in Bangladesh? How did/could these results inform policy or programming?

Reviewer #2: Despite being a well planned study there are few places to revise in the manuscript. 

Line 122-125 : might need revision. As these statements praise the work of authors-affiliated organizations. 

Line: 157-162 : The meeting in person by public health officials (DG) with agencies might be a procedure that does not need to be recalled in scientific article. It is well established that coordination is vital.

Reviewer #3: This study is very relevant to informing prevention and control interventions during humanitarian crises context. It is a significant study in the field of public health emergencies and contains needed data in moving the field forward.

If the surveillance described in this study was implemented Borno, Nigeria after people fleeing Boko Haram armed insurgency were place in camps for internally displaced persons (IDPs), the 2017 cholera outbreak in one of the camps could have been prevented. For more about the failures that lead to the 2017 Borno IDP camp outbreaks, please see 1.) https://gh.bmj.com/content/5/6/e002431.abstract, and 2) https://gh.bmj.com/content/5/1/e002000.abstract.

PLOS authors have the option to publish the peer review history of their article (what does this mean?). If published, this will include your full peer review and any attached files.

Reviewer #1: No

Reviewer #2: Yes: Lila Bahadur Basnet

Reviewer #3: No
---

## [Decision Letter · Decision Letter 1]

1 Jul 2021

Dear Abu S. G. Faruque,

We are pleased to inform you that your manuscript 'Cholera outbreak in Forcibly Displaced Myanmar National (FDMN) from a small population segment in Cox’s Bazar, Bangladesh, 2019' has been provisionally accepted for publication in PLOS Neglected Tropical Diseases.

Best regards,

Vasantha kumari Neela

Associate Editor

Amy Gilbert

Deputy Editor

Reviewer's Responses to Questions

**Key Review Criteria Required for Acceptance?**

**Methods**

-Are the objectives of the study clearly articulated with a clear testable hypothesis stated?

-Is the study design appropriate to address the stated objectives?

-Is the population clearly described and appropriate for the hypothesis being tested?

-Is the sample size sufficient to ensure adequate power to address the hypothesis being tested?

-Were correct statistical analysis used to support conclusions?

-Are there concerns about ethical or regulatory requirements being met?

Reviewer #2: The methodology and objectives of the study clearly articulated.

The study design is appropriate to address the stated objectives.

The population is clearly described and appropriate.

The correct statistical analysis were used.

Reviewer #3: There is a concern I raised in the first revision that come was not addressed.

At the introduction section of re-submission line 110, it is mentioned that cholera risk exist among FDMN due to ..., decaying immunity. Please what evidence exist to support this decaying immunity?

1. When were the FDMN vaccinated?

2. What is the duration of immunity provided by OCV?

3. Are there any scientific studies that have looked into OCV immunity status among the FBMN? If yes, does the studies show declining immunity?

**Results**

-Does the analysis presented match the analysis plan?

-Are the results clearly and completely presented?

-Are the figures (Tables, Images) of sufficient quality for clarity?

Reviewer #2: the analysis were appropriately done and well presented.

Reviewer #3: (No Response)

**Conclusions**

-Are the conclusions supported by the data presented?

-Are the limitations of analysis clearly described?

-Do the authors discuss how these data can be helpful to advance our understanding of the topic under study?

-Is public health relevance addressed?

Reviewer #2: Yes.

The conclusions are supported by the data.

Reviewer #3: (No Response)

**Editorial and Data Presentation Modifications?**

Reviewer #2: (No Response)

Reviewer #3: (No Response)

**Summary and General Comments**

Reviewer #2: I don't see the citations in this article appropriately done.

The citation is done at the end of long paragraphs. In introduction section they cite 1-7 article at the end of the paragraph. This will make the readers difficult to refer to the cited articles.

Reviewer #3: (No Response)

PLOS authors have the option to publish the peer review history of their article (what does this mean?). If published, this will include your full peer review and any attached files.

Reviewer #2: No

Reviewer #3: No

---

## [Editor Report · Acceptance letter]

31 Aug 2021

Dear Dr. Faruque,

We are delighted to inform you that your manuscript, "Cholera outbreak in Forcibly Displaced Myanmar National (FDMN) from a small population segment in Cox’s Bazar, Bangladesh, 2019," has been formally accepted for publication in PLOS Neglected Tropical Diseases.

Best regards,

Shaden Kamhawi

co-Editor-in-Chief

Paul Brindley

co-Editor-in-Chief
